# A Time Division Multiplexing Inspired Lightweight Soft Exoskeleton for Hip and Ankle Joint Assistance

**DOI:** 10.3390/mi12101150

**Published:** 2021-09-24

**Authors:** Xin Ye, Chunjie Chen, Yanguo Shi, Lingxing Chen, Zhuo Wang, Zhewen Zhang, Yida Liu, Xinyu Wu

**Affiliations:** 1Shenzhen Institute of Adanced Technology, Chinese Academy of Sciences, Shenzhen 518055, China; xin.ye@siat.ac.cn (X.Y.); lx.chen@siat.ac.cn (L.C.); zhuo.wang@siat.ac.cn (Z.W.); zhangzw@siat.ac.cn (Z.Z.); yd.liu1@siat.ac.cn (Y.L.); xy.wu@siat.ac.cn (X.W.); 2Parallel Robot and Mechatronic System Laboratory of Hebei Province, Yanshan University, Qinhuangdao 066004, China; ygshi@ysu.edu.cn; 3Guangdong Provincial Key Lab of Robotics and Intelligent System, Shenzhen Institute of Advanced Technology, Chinese Academy of Sciences, Shenzhen 518055, China; 4Shenzhen College of Adanced Technology, University of Chinese Academy of Sciences, Shenzhen 518055, China; 5Guangdong-Hong Kong-Macao Joint Laboratory of Human-Machine Intelligence-Synergy Systems, Shenzhen 518055, China

**Keywords:** lower limb assistance, time-division multiplexing, soft exoskeleton, gait event detection, iterative control

## Abstract

Exoskeleton robots are frequently applied to augment or assist the user’s natural motion. Generally, each assisted joint corresponds to at least one specific motor to ensure the independence of movement between joints. This means that as there are more joints to be assisted, more motors are required, resulting in increasing robot weight, decreasing motor utilization, and weakening exoskeleton robot assistance efficiency. To solve this problem, the design and control of a lightweight soft exoskeleton that assists hip-plantar flexion of both legs in different phases during a gait cycle with only one motor is presented in this paper. Inspired by time-division multiplexing and the symmetry of walking motion, an actuation scheme that uses different time-periods of the same motor to transfer different forces to different joints is formulated. An automatic winding device is designed to dynamically change the loading path of the assistive force at different phases of the gait cycle. The system is designed to assist hip flexion and plantar flexion of both legs with only one motor, since there is no overlap between the hip flexion movement and the toe-offs movement of the separate legs during walking. The weight of the whole system is only 2.24 kg. PD iterative control is accomplished by an algorithm that utilizes IMUs attached on the thigh recognizing the maximum hip extension angle to characterize toe-offs indirectly, and two load cells to monitor the cable tension. In the study of six subjects, muscle fatigue of the rectus femoris, vastus lateralis, gastrocnemius and soleus decreased by an average of 14.69%, 6.66%, 17.71%, and 8.15%, respectively, compared to scenarios without an exoskeleton.

## 1. Introduction

Since at least the 1890s, a variety of lower-limb exoskeletons have been designed to help normal people improve their exercise performance [1,2,3], or to help disabled people regain the ability to walk independently [4]. As a type of portable gait assistance equipment, some factors that influence the effectiveness of gait assistance need to be discussed during the design process of exoskeletons, such as supportability, portability, stiffness of exoskeleton, comfort, biomechanical characteristics of human movement, and so on. Most lower extremity exoskeletons have a rigid structure with great supportability, and have been successfully applied to the rehabilitation of incapacitated patients [5]. However, portability and comfort are difficult to guarantee because rigid exoskeletons are often bulky and heavy. Meanwhile, due to the large inertia of the rigid structure and corresponding joint misalignments, a powerful actuator and a complex sensing system are required for the exoskeleton to mimic and assist biological joint motion accurately. Therefore, the soft exoskeleton, which avoids the disadvantages of a rigid exoskeleton, has attracted the attention of many scholars [6]. However, whether the exoskeleton is soft or rigid, the weight of the exoskeleton will partly counteract efforts of the assistance, because extra mass changes the walking gait pattern [7]. Reducing system mass often improves portability and user comfort, and reduces additional inertia. It is necessary to reduce the weight of the device without compromising its effectiveness.

Generally, there are two methods to reduce a system’s weight. On the one hand, we can remove unnecessary structural parts or substitute with lighter materials to manufacture the structure of the equipment to make the system more compact and lightweight. It may work significantly for traditional rigid exoskeletons where most of the structural parts are made of metal [8,9]. However, where soft exoskeletons are concerned, it is not obvious since most of its structural parts are made of soft textile fabrics and lightweight plastic. On the other hand, we can remove the number of drive parts, which are large and heavy, to essentially reduce device weight. To ensure the independence of joint movement, at least one motor is required for each joint so that the exoskeleton can assist all kinds of body movements. Therefore, a large number of motors is required as the number of joints increases. A lightweight, portable, soft exosuit with two motors for one ankle joint has been proposed to assist paretic plantar flexion and dorsiflexion during walking for people who suffered from a stroke in [10]. Zhou et al. [11] designed a novel soft knee exoskeleton based on a continuum structure to assist knee extension with one motor for one leg. Ye et al. [12] presented a reconfigurable multi-joint actuation platform that is capable of providing biological torques to the ankle, knee, and hip joints. To reduce the mass of systems, it is an effective method to utilize the clockwise and counterclockwise rotation of the motor to realize the different movement. For example, the flexion and extension of the same joint are assisted by single motors in [13,14,15], or one motor to be used for hip flexion and knee extension on the same leg [16], or one motor to be used for the same joint on both legs, such as right ankle plantar flexion and left ankle plantar flexion [17]. However, as the walking speed increases, the assistive force profile becomes shorter or the peak decreases, because there is less time left for the motor to change direction of rotation. In addition, changing the power source or removing the power of the exoskeleton may also be a method to reduce the weight of the system. Wang et al. [18] designed a modular soft-rigid pneumatic lower limb exoskeleton that assists the hip, knee, and ankle joints pneumatically, which proves that one actuator is capable of driving all six joints of the lower limbs. Pneumatic muscles mimic the strength of human muscles by contracting length, but often in the same amount as the number of joints that need to be assisted [19,20,21,22]. The unpowered exoskeletons always have little mass but do not provide much assistance [23,24].

Meanwhile, in order to deliver the force accurately to the target joint, some critical gait events must be captured by the sensing system for feedback control. Asbeck et al. [25] employed the foot switches to detect heel strike and they combined a gyroscope in the heel with a tension sensor to detect heel strikes and toe-offs in other work [17]. Karavas et al. [12] mounted three IMUs on each leg to obtain sagittal-plane angular displacement and angular velocity of the thigh, shank, and foot, respectively.

In this paper, a lightweight soft exoskeleton inspired by time-division multiplexing was proposed. As shown in the Figure 4 in Section 2.3, it utilizes only one motor to assist hip flexion and ankle plantar flexion of both legs at the corresponding phases during a gait cycle. A lightweight device named an automatic winder was designed to change the force loading path dynamically during walking. In addition, a new gait event detection method that detects toe-off events and the kinematic information of the hip joint though a pair of IMUs mounted on the thighs was implemented. The system design of the soft exoskeleton and its motion characteristics are analyzed in detail in Section 2. The assistance strategy, motion detection method, and control algorithm are described in Section 3. The results of muscle fatigue experiments are presented in Section 4. Our soft exoskeleton is discussed in Section 5. Finally, a conclusion is drawn in Section 6.

## 2. System Overview

The soft exoskeleton designed in this work can provide assistive force to hip joint flexion and also to ankle plantar flexion. It is used mainly to reduce muscle fatigue of human walking. Instead of each force path being controlled by individual motor, multiple force paths are totally controlled by only one motor through the automatic winder. In order to minimize the weight of the system, in this work, a soft exoskeleton is designed in a highly integrated manner.

### 2.1. Time Division Multiplexing and Joint Synergies

Time division multiplexing (TMD) is a concept in communication engineering that is used for transmission of digital signals. TMD uses different time-periods of the same physical connection to transmit different signals, thus achieving the purpose of signal multiplexing. Providing assistance to the body during walking and running is to transfer different assistive forces to different joints through exoskeleton, which is similar to the process of transmission of multiple signals. Meanwhile, although the biological moment curve is continuous, the key motions are essentially separable on the time-domain during walking and running. Therefore, inspired by TMD, the movements during walking that do not overlap on the timeline were selected to assist in this work, and the gait cycle was divided into independent time segments according to these movements. In different time segments, the same motor provides assistive forces to different joints. The actuation scheme is achieved through the opposite rotation of one motor and automatic winders described below.

Except for some special motions, most of the lower limb movements in humans are periodic and symmetrical, and the legs are 180 degrees out of phase. For example, when walking, except for the double limb stance phase, when one leg is in the stance phase, the other leg must be in the swing phase; in running, the movement of both legs is completely symmetrical with no overlap. In addition, some joint motions are discontinuous on the timeline, which makes it possible to control two motions with the opposite rotation of one motor, such as one motor for the same joint on both motions (e.g., hip flexion and hip extension), or one motor for the same joint on both legs (e.g., right ankle plantarflexion and left ankle plantarflexion) [17]. Some joint motions are continuous on the timeline (e.g., hip flexion and ankle plantar flexion of the same leg), which makes it impossible to control both motions with a single motor because there is no time for the motor to reverse their direction.

A lightweight device named an automatic winder is designed to achieve the control of two continuous motions on the timeline with the single motor by changing the loading path dynamically during the process of movement, shown in Figure 1a. The cable is wound around the spool, inside of which the spiral torsion spring is placed. When the cable is pulled out of the shell, the spool drives the spiral torsion spring to contract, to store the elastic potential energy. When the cable is released, the spiral torsion spring releases the elastic potential energy and drives the reverse rotation of the spool to automatically recover the cable. One end of the coupling is fixed with the spool and the other end of the coupling is fixed with a brake wheel. The brake wheel engages with the brake pad through the spline. The electromagnet is fixed with shells which are the gray parts in Figure 1a. When the power is on, the brake pad is sucked up by the electromagnet, leaving the brake wheel and the spools fixedly attached to it in a self-latching state (the cable cannot be retracted or pulled out). When the power is off, the electromagnet loses its magnetism, and the break pad is separated from the electromagnet, so that the brake wheel and spool is in a free state (the cable can be freely pulled out or retracted). The peak force of the automatic winder in the self-latching state reaches 100 N. An automatic winder was placed at the end of the loading path, shown in Figure 1b. The cable of the automatic winder A was connected to the actuator to form the hip flexion force loading path MA. The cable of the automatic winder B was attached to point P on the loading path MA. During the toe-offs movement process, the automatic winder A was turned off and B was turned on, to transfer the force to point B along the loading path MB, instead of to point A along MA. Before the toe-offs phase is completed, winder A is turned on, and the loading path MA starts to preload; when the toe-offs phase ends, the winder B is cut off, and the force-loading path is changed from MB to MA completely. Without motor stop rotating, assistive force is provided to two separate joints in sequence from the single motor.

Combined with the rotation of both the direction of the motor and the mechanism of the automatic winder to dynamically change the force loading path, one motor is able to output four force loading paths, shown in Figure 1c. The pully of the motor is winded in both directions, and the two cables in the opposite direction pass through the automatic winder, respectively, resulting in four force loading paths that can be dynamically switched.

The spiral torsion spring is tightened several rotations to store elastic potential energy in advance during assembly, so that the preload of 3–5 N is applied to the force loading paths PA and PB. Therefore, a preload of 6–10 N is applied to path WP. The preload makes the force loading path always in a tensioning state when the system is working, which is equivalent to improving the stiffness of the system, and then the assistive force can be applied to the target joint faster, and the cable cannot jam between the bidirectional pully and case. The torque exerted on the hip and ankle joints resulting from the preload is 0.24–0.32 Nm and 0.12–0.20 Nm, respectively, which are too small to be felt by subjects (the mean hip rotation radius is 0.08 m and the mean ankle rotation radius is 0.04 m).

### 2.2. System Calculations

In the movement of human lower limbs, the flexion and extension of the three joints of each leg can be regarded as a total of six movements from the sagittal plane, and the push-off movement expends much of the energy during walking [26]. Therefore, it is efficient to provide assistance to the toe-offs phase. When selecting the other motions to assist, we ensure that all motions do not overlap in phase so the time required for the motor to change the direction of rotation is sufficient. In terms of walking, it is hard to assist hip extension and ankle plantarflexion at the same time due to the toe-offs movement of the one leg and the hip extension movement of the other leg overlapping in the double limb stance phase, shown in Figure 2. As for running, a combination of hip extension and plantar flexion is most preferred because there is no overlap in phase between the two legs and there is a period of empty phase left for the motor to change the direction of rotation. Regardless of whether it is walking or running, there is no overlap between the hip flexion movement and the toe-offs movement of the separate legs. Therefore, we choose the hip flexion and toe-offs movement of both legs to assist. The arrangement of the actuator, automatic winder, and force loading path is shown in Figure 1c.

We connected the cable of the automatic winder in the ankle suit to point P on the loading path WA to form the loading path WB. Therefore, the first thing to do for a force loading path is to determine the location of point P. A simple geometric model of the wearer’s leg including the cable attachment point is shown in Figure 3. Force is provided along the loading path when the motor is retracted, which causes point P to move towards point W. If the system is completely rigid, the amount of cable the motor pulls in each time is equal to the amount of shortening of the load path caused by the body changing position ($\Delta g_{hipflex}$ and $\Delta g_{pushoff}$). When the stiffness of the system is taken into account, a part of the length of the winded cable offsets the deformation of each part, including the waist belt ($\Delta s_{waist}$), the thigh brace ($\Delta s_{thigh}$), the calf brace ($\Delta s_{calf}$, $\Delta s_{knee}$), and the ankle suit ($\Delta s_{ankle}$). The actuator position is defined as the required cable length pulled by the actuator (*l*), given by Equation (Equation 1). The tiny deformation of the cable under stress is ignored.

Because two cables are coiled around the pully on the motor in opposite directions, when the actuator pulls the cable in one direction, the cable of the same length will be released in the other direction which makes point P’ move towards point A’. Point P is set at the midpoint of the WA because it cannot enter the Bowden sheath and the automatic winder due to structural limitations. For this reason, Equation (Equation 2) is given for the length relationship, and the cable length pulled by the actuator should be less than half of WA.
(1)$$l=\Delta g_{hipflex}+\Delta g_{pushoff}+\Delta s_{ankle}+\Delta s_{waist}+\Delta s_{calf}+\Delta s_{knee}$$
(2)$$l<\frac{\bar{WA}}{2}$$

### 2.3. Ankle Suit and Hip Suit

The specific suit discussed and evaluated in this paper is shown in Figure 4a. The hip suit is composed of a waist belt and thigh braces, and the automatic winder A is securely attached to the thigh brace. The ankle suit is composed of a foot attachment suit and a calf brace. To guarantee that point P does not go beyond the path WA, as in Equations (1) and (2), the path WA is desired as long as possible and the stiffness of the suit must be maximized to minimize the deformation of the suit. For this reason, the attachment point W and the automatic winder A, B is located at the superior iliac spine, the upper femoral head of the thigh, and the heel of the shoe, respectively, because these three locations have the least muscle tissue and maximum stiffness. One end of the Boden cable sheath of the calf brace is fixed to the center of rotation of the knee joint to prevent interference torque to the knee joint, and the other end is located at the most prominent point of the calf gastrocnemius muscle. In addition, the average length of the WA is 46 cm, tested by six subjects (age 24 ± 3 year, mass 70.0 ± 8 kg, height 1.75 ± 5 cm). Initial experiments indicated that the pull amplitudes of 5–6 cm are necessary to create 100 N in the loading path of MA and MB, and 1–2 cm is required for both loading paths to achieve their tension position. In addition, the sum of normal working $\Delta g_{hipflex}$ and $\Delta g_{pushoff}$ is 7–9 cm. Thus, the total pull amplitude l is 15–22.5 cm, less than WA/2 tested in this work.

To deliver the assistive force to the body accurately, it is necessary to establish the suit-human series stiffness model due to the exosuit system being highly nonlinear. Six subjects (age 24 ± 3 year, mass 70.0 ± 8 kg, height 1.75 ± 5 cm) were asked to stand on the flat with a 45 cm distance between their feet, as shown in Figure 4a. Automatic winder A is turned on and automatic winder B is turned off during the stiffness experiment of the hip suit so that the force is applied to the thigh brace. The actuator retracts the Boden cable at 5 mm/s and stops when the force is greater than 100 N. The stiffness model of the hip suit for hip flexion is shown in Figure 4c. Following the same procedure as with the hip suit, automatic winder A is turned off and automatic winder B is turned on during the stiffness experiment of the ankle suit so that the force is applied to the ankle suit. The stiffness model of the ankle suit for ankle plantar flexion is shown in Figure 4b.

### 2.4. Actuation System and Sensors

Based on the requirements described above, an actuator with single motor has been designed as shown in Figure 4a. In this actuator, a Brushless Motor (M2006, DJI, Shenzhen, China) connected directly to a planetary gearbox of ratio of 36:1 to drive an aluminum alloy bidirectional pulley having a diameter of 25 mm. The bidirectional pully needs more than three default complete rounds of cable in both rotation directions to ensure that when the cable is drawn in one direction, the other cable is retracted in the other direction. Without default rounds of cable, when the motor rotates, both cables will be drawn into the pully. An ESC (C610, DJI, Shenzhen, China) which communicates with a MCU (STM32F407, STMicroelectronics, Milano, Italy) through the CAN-Open communication protocol connects the motor to an electronics board.

There are two load cells and two inertial measurement units (IMUs) in total to collect information for system control. One end of each load cell is connected to the end of the cable from the actuator, and the other end is connected to the end of the cable from the automatic winders so that the change of force on the suit can be monitored in real-time. Additionally, two IMUs (LPMS-B2, ALUBI, Guangzhou, China) are mounted on the thigh brace and the IMU measurements are used in the gait detection algorithm (see Section 3.2). Finally, 62% of the weight of our soft exoskeleton is put on the waist, which is close to the center of mass of the body, in order to minimize the additional energy cost due to the weight of the exoskeleton [27]. The total weight of our exoskeleton is 2.24 kg. The weight of each part is given in Table 1.

## 3. Control

### 3.1. Assistance Strategy

The moments and power of flexion and extension in the lower limb joint are analyzed in [28,29,30,31]. As depicted in Figure 5a, the red chain dotted line and green chain dotted line show the required biological torque for the ankle and hip joints, respectively, during walking for a gait cycle, starting with the heel strike of one leg to the next heel strike of the same leg. Ding et al. [12] proposed a reasonable assistive force profile that takes advantage of the overlap of the biological torques of ankle plantar flexion and hip flexion to assist both movements simultaneously by using only one actuator, as shown by the red solid line in Figure 5a. To prevent interference with the natural movement of the ankle, the force must be removed immediately at the end of the toe-offs phase, which represents the end of the plantarflexion movement. However, the hip flexion does not end with the toe-offs. From the functional event diagram of the gait cycle, it can be observed that the hip flexion moment is mainly used to balance the gravity of the swing leg to maintain the velocity of the swing leg when the toe is pushed off and the swing leg enters the initial stage of the swing. Although the hip flexion torque applied to the swing phase is small compared to the terminal stance and pre-swing phase, its power is large due to the high speed of the leg at the initial swing. After the toe-offs, the power continues to increase to the peak and then decreases, as shown in Figure 5b. Meanwhile, as the walking speed increases, the time of peak power of hip flexion will be further and further away from the time of toe-offs [28,31,32]. For these reasons, it is inefficient to end the assistance of hip flexion at the ending of toe-offs. As shown in Figure 4a, taking the left leg as the observation object, with automatic winder A in a free state and the automatic winder B in a self-latching state between 18% and 64% of the gait cycle, assistive force is delivered to the ankle joint in this period. Before the toe-offs end, winder A is turned on and the loading path MB starts to preload; when the toe-offs end, the winder B is cut off, and the force loading path is changed from MB to MA completely. At maximum hip flexion angular velocity, which means the left leg starts to slow down in preparation for the next heel hit, the winder A is powered off, ending the assistance for hip flexion.

For the leg in the stance phase, the relationship between the biological torques of each joint and the ground reaction forces is relatively complicated. For the purposes of safety and not interfering with the natural movement of the human body, the force profiles for actuating the plantar flexion was simply scaled to 5% of the biological torque, as shown in Figure 5a. However, for the leg in the swing phase, each joint moves independently, and the swing speed of this leg is mainly controlled by the muscles of the hip joint, so the assistive force can be properly improved without affecting the natural movement of the human body. The force profile for actuating the hip flexion after the end of the toe-offs is finally scaled to 25% of the biological torque, as shown in Figure 5a. The final force profile applying to the hip flexion is a composite of the indirect moment curve of the ankle joint and the direct moment curve of the hip joint after the end of the assistance to the ankle.

In this work, the assistive force profiles discussed above are fitted in a sinusoidal curve, as shown in Figure 5c. For the six subjects, the mean maximum biological moment of ankle plantar flexion is 60 Nm and the mean ankle rotation radius is 0.04 m, which means the peak assistive force applied to the ankle is 80 N; the mean maximum biological moment of hip flexion after the end of toe-offs is 20 Nm and the mean hip rotation radius is 0.08 m, which means the peak assistive force applied to the hip after the end of toe-offs is 62.5 N.

When the power is switched from one leg to the other, it takes some time for the motor to change its direction of rotation. Therefore, the onset of ankle power is delayed from 18% to 38% of the gait cycle. Meanwhile, the delay of the start time does not significantly affect the assisting effect because the ankle almost does not provide positive work before 38% of the gait cycle.

### 3.2. Gait Event Estimation Using Imu

In order to transfer the force accurately to the target joint, a new gait event detection method is implemented, which detects the toe-offs event and the kinematic information of the hip joint through a pair of IMUs mounted on the thighs. This method is different from other methods [12,17,25] since it does not detect toe-offs directly through the ankle-mounted IMUs or foot switches, but recognizes the maximum hip extension angle to characterize toe-offs indirectly.

To develop this new method, a portable lower limb movement information collection platform consisting of five IMUs and a pair of plantar pressure sensors was constructed to measure the kinematic information of lower limbs and ground reaction forces during walking and running in a variety of terrain, as shown in Figure 6a. By collecting and observing the experimental data when the subjects (age 24 ± 3 year, mass 70.0 ± 5 kg, height 1.75 ± 5 cm) were walking on the treadmill at 4.5 km/h, as shown in Figure 6b,c, pressure builds up on the forefoot and increased hip extension puts the limb in a more trailing position at the terminal stance, and when the top tip pressure reaches its maximum, the hip extension angle also reaches its maximum. Based on these observations, a new gait event detection method focused on capturing the maximum hip extension angle which can be used to indicate the toe-offs was applied to accurately transfer forces to the ankle. Meanwhile, the maximum hip flexion velocity which means the end of positive work of hip flexion during the swing period is detected as the end time of providing assisted force to the hip flexion.

In conclusion, an IMU was mounted at the center and front of the thigh brace on each leg to obtain the angle and angular velocity of the thigh. The timing of the maximum hip extension angle and the maximum hip flexion velocity were captured to represent the moment of toe-offs and the end of the positive work of hip flexion, respectively.

### 3.3. Controller Design

Considering the unpredictable changes in system parameters due to the braces’ migration and other variabilities during walking, although the stiffness of system was improved by designing the braces with as much stiffness as possible and applying a pre-tightening force through an automatic winder, it was hard to deliver assistive force to the desired joints accurately. To offset the errors caused by all kinds of variabilities, the PD iterative controller was designed in this work.

The control architecture is depicted in Figure 7. Through the gait event detection algorithm mentioned above, the timing of toe-offs and the timing of maximum hip flexion are captured by IMUs attached to both thighs, which represents the timing of peak amplitude of the ankle plantar flexion assistive force profile and the timing of the end of hip flexion assistive force profile, respectively. In addition, the gait cycle (GC) of the next iteration was obtained through the weighted average of the previous three gait cycles, as shown in Equation (Equation 3), and the weighting coefficients were obtained experimentally.

Meanwhile, two load cells placed in series with the cable of actuator and automatic winders at thigh and ankle collect real-time forces to extract key features, such as onset timing, end timing, and peak force amplitude were applied to the hip and ankle. Comparing the desired force profile and the actual force profile of the current gait cycle, we adjusted the desired force profile of the next gait cycle to better transmit the assistive force. However, the iterative control is an open-loop system without the function of eliminating real-time errors. Therefore, a proportional-derivative (PD) controller was added as a feedback controller, which is expressed by Equation (Equation 3).
(3)$$P_{t}=K_{p}e_{F}+K_{d}\Delta e_{F},$$
where $K_{p}$ϵ$R^{2\times 2}$ and $K_{d}$ϵ$R^{2\times 2}$ denote the coefficient matrices of the proportional and derivative controllers, respectively.

## 4. Experimentation

During walking, muscle fatigue increases over time, causing a loss of muscle strength, which frequently limits the performance of motor tasks. Gait appears most robust to weakness of hip and knee extensors and it is most sensitive to weakness of ankle plantar flexors, hip abductors, and hip flexors [33]. The soft exoskeleton proposed in this work was used to provide assistance to ankle plantar flexors and hip flexors to improve people’s athletic endurance. Therefore, the performance of our lightweight soft exoskeleton was evaluated through the muscle fatigue experiment. Six adult men (age 24 ± 3 year, mass 70.0 ± 8 kg, height 1.75 ± 5 cm) without any musculoskeletal injuries or disorders were recruited to participate in the experiment. All participants provided written informed consent prior to participation in the study. The experiment was approved by the Medical Ethics Committee of the Shenzhen Institute of Advanced Technology ((SIAT)-IRB-200715-H0512 (valid time from 2020.01 to 2022.12)), and the content of the experiment and its possible impact were explained to all the participants before conducting the experiment.

### 4.1. Muscle Fatigue Experiment

As shown in Figure 8, the muscle fatigue experimental equipment consisted of a host computer, *sEMG* (Surface Electromyography with a sampling frequency of 2000 Hz), treadmill, and lightweight soft exosuit. The *sEMG* electrode was placed on the right leg, and four muscles were examined: the rectus femoris (RF), vastus lateralis (VL), gastrocnemius (GAS), and soleus (SOL). The *sEMG* signals were collected while the subjects were walking on a treadmill at a speed of 4.5 km/h. There were three muscle fatigue tests of the lightweight soft exosuit to be completed, which included the static steady state test, normal walking test without the exosuit, and normal walking test with the exosuit.

#### 4.1.1. Static Steady-State Test

The subjects wearing the *sEMG* device were asked to stand still on ground that was horizontal for 1 min with hands naturally hanging down, and the *sEMG* signals were then collected during standing.

#### 4.1.2. Normal Walking Test without Exosuit

The subjects wearing the *sEMG* device were asked to walk on a treadmill. It took a while for the subjects to stabilize their gait when they were moving at a high speed on the treadmill suddenly. This affected the accuracy of the *sEMG* data. In order to obtain a stable walking gait, the walking speed gradually increased from 2 km/h to 4.5 km/h, rising by 0.5 km/h each time; and when the speed reached 4.5 km/h, the subjects continued walking for 5 min. The *sEMG* signals were then collected during stable walking at a speed of 4.5 km/h.

#### 4.1.3. Normal Walking Test with Exosuit

Similarly to the normal walking test without the exosuit, the subjects wearing the *sEMG* device and our lightweight exosuit were asked to walk on a treadmill and the walking speed gradually increased from 2 km/h to 4.5 km/h, rising by 0.5 km/h each time; and when the speed reached 4.5 km/h, the subjects continued walking for 5 min. During the walking process, the exosuit provided assistive force to the hip joints and ankle joints, respectively. The *sEMG* signals were then collected during stable walking at a speed of 4.5 km/h.

To give the muscles a full rest, there was a 20-min break between experiments and all the muscle fatigue experiments were required to be completed on the same day because the experiments were not intense.

In this research, the rectus femoris (RF), vastus lateralis (VL), gastrocnemius (GAS), and soleus (SOL) which are commonly used for normal walking [34,35] were used as the experimental muscles. For the results of the muscle fatigue experiments, we used root mean square (*RMS*) as the evaluation indicator of the degree of muscle fatigue:(4)$$EMG_{RMS}\left(n\right)=\sqrt{\frac{1}{n}\sum_{i=n-N+1}^{n}sEMG{\left(i\right)}^{2}},$$
where *sEMG(i)* is the *sEMG* signal of each muscle.

The level of muscle fatigue was analyzed by the value of *RMS* of the *sEMG* signal, and the lower *RMS* value means the lower level of muscle fatigue [36]. The reduction of muscle fatigue of the subjects is shown in Table 2. During human walking, the calf muscles contract concentrically at approximately 38–62% in the gait cycle, which includes the phase of Terminal Stance and the phase of Pre-Swing [30], to provide power to complete ankle plantar flexion. At this period, the automatic winder attached at the ankle frame was turned on, while the automatic winder attached at the thigh brace was turned off, so that the assistive force from actuation is delivered to point B while only a preload of 3–5 N from the automatic winder is applied to point A, as shown in Figure 3a. The soleus muscle provides power mainly during walking, and the gastrocnemius muscle provides energy mainly during the sit-to-stand transition or jumping [34,35], which means the assistance for ankle plantar flexion is more beneficial to the soleus muscle than to the gastrocnemius muscle during walking. As illustrated by Table 2, all but the second subject showed a significant reduction in soleus fatigue, which is distinctly higher than the reduction in gastrocnemius fatigue, compared to normal walking without the exosuit. At the hip, the rectus femoris and other hip flexors contracted concentrically at approximately 50–72% in the gait cycle, which includes the phase of Pre-Swing and the phase of the Initial Swing [30]. Because of the synergy between the ankle plantar flexion and hip flexion, the assistive force applied to the ankle joint was also delivered to the hip joint at 50–62% in the gait cycle, as shown in Figure 5a. After the end of ankle plantar flexion, the automatic winder attached to the ankle frame was turned off, while the automatic winder attached at the thigh brace was turned on, so that the assistive force from actuation was delivered to point A while only the preload of 3–5 N from the automatic winder was applied to point B, as shown in Figure 3b. The reduction of rectus femoris fatigue is second only to that of the soleus muscle, as shown in Table 2. Meanwhile, due to the subtle difference in the way each person walks, the level of muscle fatigue decreased in different patterns with the exosuit.

In order to clearly visualize the amount of reduction in muscle fatigue when wearing the exosuit, a bar chart comparing the difference is shown in Figure 9. It can be found that compared with normal walking without the exosuit, the level of muscle fatigue of the RF, VL, GAS, and SOL of normal walking with the exosuit was reduced by 14.69%, 6.66%, 8.15%, and 17.71%, respectively. The results demonstrate that using the lightweight soft exosuit could reduce the level of muscle fatigue to some extent.

## 5. Discussion

This work presented a body-worn time division multiplexing inspired lightweight soft exoskeleton, which provides assistance for hip flexion and ankle flexion. We designed the automatic winder that dynamically changes the force loading path in the process of movement. Combined with the characteristics of the automatic winder and the the clockwise and counterclockwise rotation of the motor of the motor, it realizes the use of a single motor to assist the movement of the four joints of two legs (hip flexion and plantar flexion for both legs) which makes the weight of the actuator significantly reduced compared to prior designs. In addition, the timing of the maximum hip extension angle and the maximum hip flexion velocity were captured to represent the moment of toe-offs and the end of the positive work of hip flexion, respectively. The effectiveness of the lightweight soft exoskeleton was evaluated by muscle fatigue experiments.

The comparison of some typical soft exoskeletons in various assistance methodologies is presented in Table 3. Walking or running is accomplished mainly through the flexion and extension of the hips, knees, and ankles of the lower limbs in an orderly arrangement of the timeline. Therefore, assisting any joint movement at an appropriate time can enhance the athletic ability of the human body [10,11,37,38]. When assisting more than two joints, the simplest solution is to increase the number of motors and arrange the corresponding force loading path, but this can significantly increase the weight of the system [39]. Some papers suggest that the symmetry of human lower limb movement can be used to reduce the number of motors. Asbeck et al. [17] used the clockwise and counterclockwise rotation of the motor to assist the same joint in both legs, while Zhang et al. [16] used the same method to assist two joints in the same leg. In this paper, we took advantage of the symmetry of lower limb movement and combined it with the automatic winder, dynamically changing the force loading path to achieve the goal of a single motor assisting the hip flexion and plantar flexion in both legs. Meanwhile, the problem of the cable being loosened in the opposite direction when the motor rotates was solved by applying preload force on the force loading path by the automatic wire winder. With a total weight of 2.24 kg, the lightweight exosuit in this paper is the lightest soft exoskeleton known to assist the four joints.

Despite this idea being novel, hip flexion and plantar flexion in both legs were actuated by a single motor, and the weight of the exosuit was significantly reduced. There are, however, some important questions that need to be discussed. Firstly, although the number of motors was reduced, four automatic winders were added, each weighing 55 g. This means that the added mass must be less than the mass of the motors which were subtracted. As shown in Equation (Equation 5), when the motor is paired with the biological joint, the actuation scheme proposed in this paper is better if $M_{single}$ is greater than 73 g. When the motor is used to assist two joints [16,17], the actuation scheme proposed in this paper is better if $M_{single}$ is greater than 220 g. Secondly, the maximum hip extension angle used to represent the toe-offs was only suitable for flat walking, and was not tested in other terrains. Thirdly, as walking speed increases, it is possible to reduce the peak force or shorten the assistance force interval, due to a shorter time being left for the motor to reverse, which may weaken the soft exoskeleton’s ability to reduce muscle fatigue. The maximum speed without changing the force profile was not studied in this work. In addition, the actuation scheme proposed in this research focused on symmetric gait, and cannot be applied to assist asymmetric gait. This means that the occasional asymmetrical gait during walking can diminish the effects of walking assistance. Finally, we were limited to the constant speed of the treadmill, but in fact, the speed of people walking is arbitrary. The actuation scheme for different speeds will be carried out in later work.
(5)$$\left\{\begin{array}{cc} 3M_{single}-M_{total}>0 & (a)\;One\;motor\;for\;one\;joint\\ M_{single}-M_{total}>0 & \;\;(b)\;One\;motor\;for\;two\;joints \end{array}\right.$$

## 6. Conclusions

This work presented a novel soft exosuit which assists hip flexion and ankle planter flexion in both legs with a single motor which outputs power to the body throughout the gait cycle. A lightweight device named the automatic winder was designed and installed at the end of the Boden cable to dynamically change the force loading path and apply the pre-tightening force on the Boden cable, increasing the stiffness of the system to some extent. The relationship between plantar flexion movement and hip flexion movement in the time domain was fully analyzed, and the composite force profile of hip flexion movement was obtained. The toe-offs event was represented by the maximum hip extension angle obtained from the IMU attached to the thigh. The muscle fatigue experiments showed that using the lightweight soft exoskeleton can reduce muscle fatigue by about 14.69%, 6.66%, 8.15%, and 17.71% for the rectus femoris, vastus lateralis, gastrocnemius, and soleus, respectively. In summary, this study focused on reducing muscle fatigue of human walking and essentially reducing the weight of the whole system. An actuation scheme in which a motor assists the motions of four separate joints was proposed through exploring the relationship between the movements of the lower limb joints. This also provides a new perspective of soft exoskeleton design, to engage from the entire gait cycle rather than individual joint movements. In the future, we intend to study the force profile at different speeds, the provision of more accurate assistive force applications to the joints, and a suitable reducer being added to the automatic winder to increase its maximum load capacity.

## Figures and Tables

**Figure 1 micromachines-12-01150-f001:**
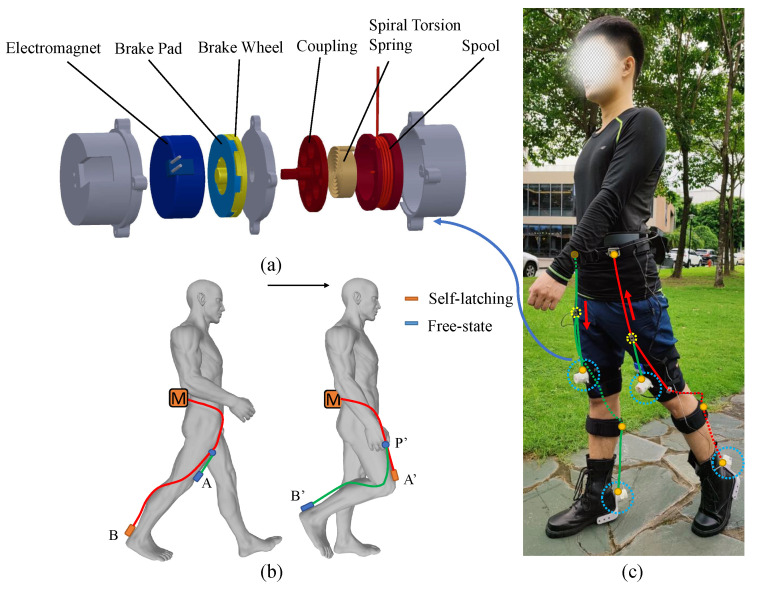
(**a**) A 3D CAD diagram of the actuation system. (**b**) Schematic diagram of automatic winder to change the force loading path from PB to PA dynamically. (**c**) Photograph of the system in use during walking outside. Actuators are mounted on the waist and the working path is marked red and the slack path is marked green.

**Figure 2 micromachines-12-01150-f002:**
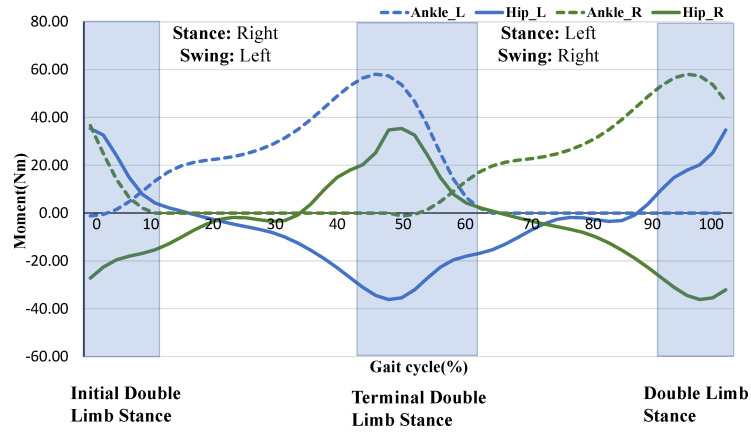
Biological moment curves of the hip and ankle joints of both legs during walking. The dashed blue line represents the biological moment curve of the right ankle and the solid blue line represents the hip. The dashed green line represents the biological moment curve of the right ankle and the solid green line represents the hip.

**Figure 3 micromachines-12-01150-f003:**
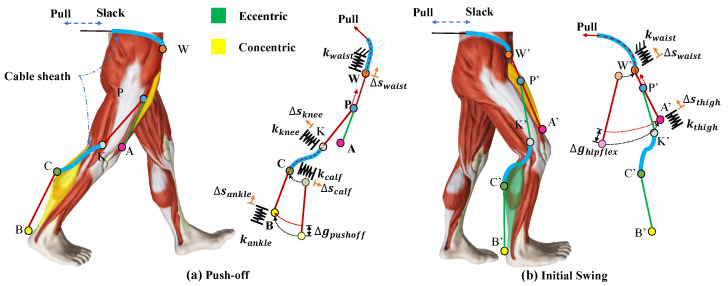
Sequence of muscle activation from toe-offs to initial swing and a simplified spring model of the human-exosuit interface. (**a**) Represents the timing of toe-offs. (**b**) Represents the phase of initial swing. The active muscles are highlighted, with the color indicating whether the muscles are shortening (concentric contraction) or lengthening (eccentric contraction) and the brightness indicating the magnitude of the muscle activity.

**Figure 4 micromachines-12-01150-f004:**
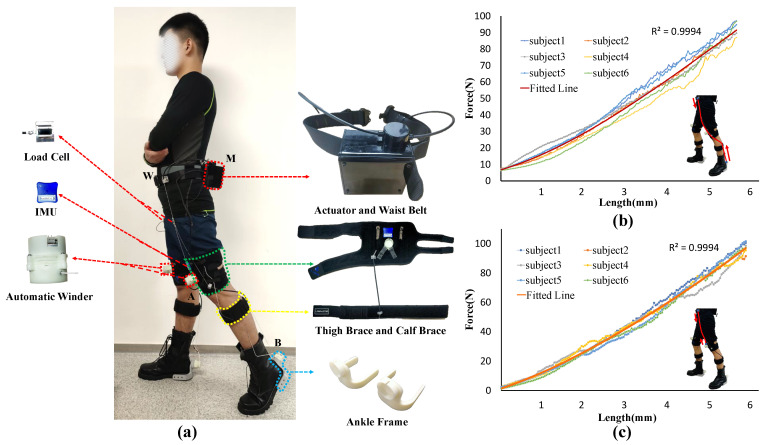
(**a**) Exosuit hardware components. (**b**) The stiffness model of the ankle suit. (**c**) The stiffness model of the hip suit. The experiment was conducted by retracting the Bowden cable while holding the initial standing posture.

**Figure 5 micromachines-12-01150-f005:**
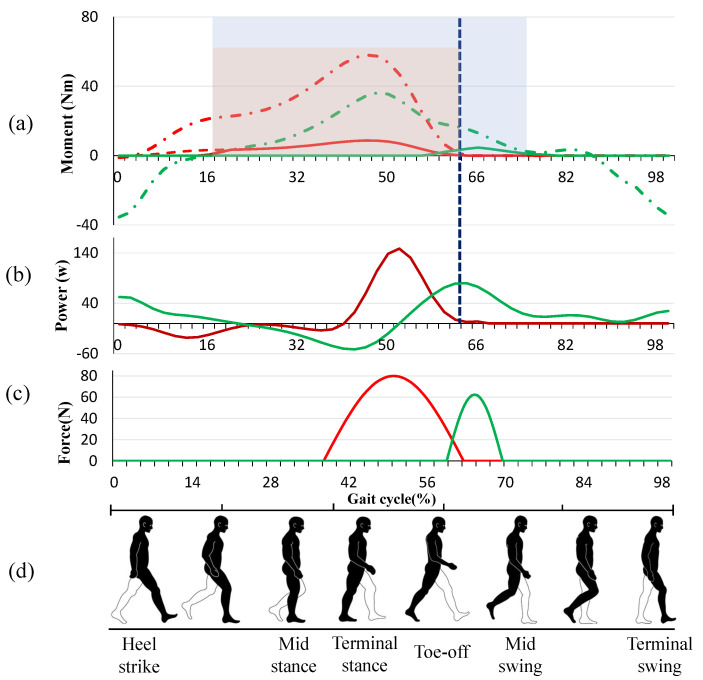
(**a**) Moment trajectory for the ankle and hip during walking. The red chain dotted line and green chain dotted line represent the required torque for the ankle and hip joints, respectively; the red solid line means the required moment applied to the ankle joint, which is 5% of the biological moment of the ankle; the green solid line means the required moment after the end of toe-offs, which is 25% of the biological moment of the hip after the end of toe-offs. The blue shaded area represents the range of assistance to the hip; the red shaded area represents the range of assistance to the ankle, contained in the blue shaded area. (**b**) The power curve of the ankle joint and hip joint during walking. The red solid line represents the ankle while the green represents the hip. (**c**) Actual force trajectory. The red solid line represents the force applied to the ankle through the automatic winder located at the ankle suit and the greed solid line represents the force applied to the hip through the automatic winder located at the thigh brace. (**d**) Human gait cycle analysis.

**Figure 6 micromachines-12-01150-f006:**
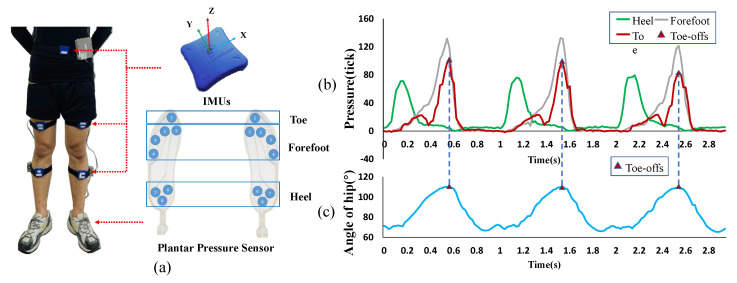
Toe-offs identified based on data from IMUs attached to the thigh. (**a**) The lower limb movement information collection platform. The sampling frequency of the system is 100 Hz. (**b**) The pressure ticks of the heel, forefoot, and toe during three cycles. Toe-offs can be clearly distinguished by foot pressure distribution. (**c**) The angle of the hip, where an increased angle indicates hip extension.

**Figure 7 micromachines-12-01150-f007:**
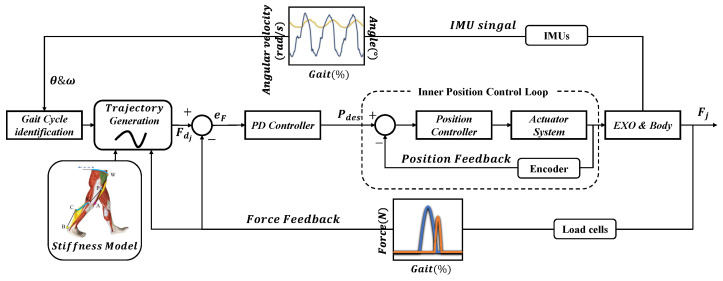
Control architecture with an iterative scheme in parallel with the PID feedback controller. The index j denotes the iteration of the algorithm, $F_{dj}$ denotes the desired force in the jth iteration, $F_{j}$ is the force collected through load cells, $e_{F}$ is the difference value of $F_{dj}$ and $F_{j}$, $P_{des}$ is the desired motor position of the controller output, θ denotes the real-time angle of the hip, and ω denotes the real-time angular velocity of the hip.

**Figure 8 micromachines-12-01150-f008:**
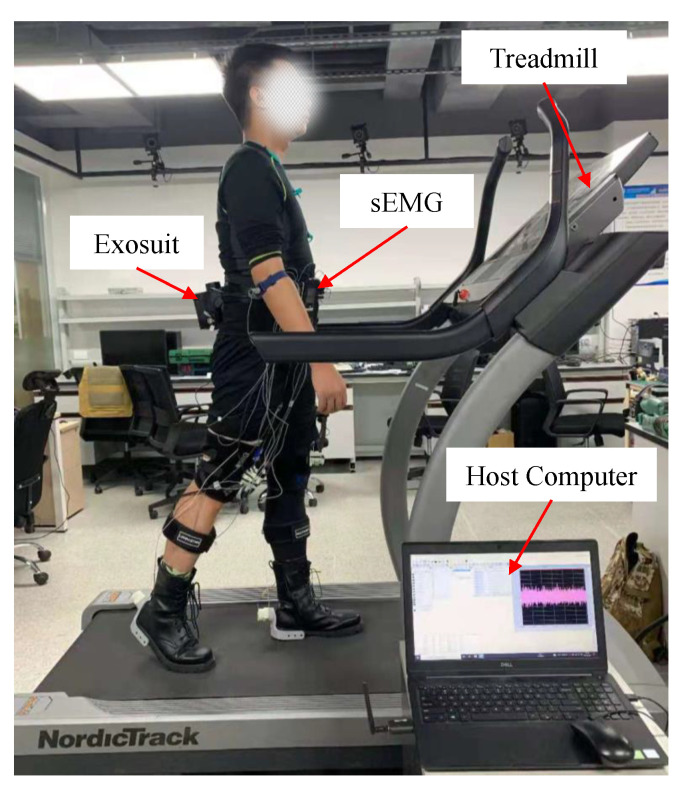
Evaluation experiment of the lightweight soft exoskeleton with *sEMG* device. Subjects wear the lightweight exosuit and measure the muscle fatigue degree.

**Figure 9 micromachines-12-01150-f009:**
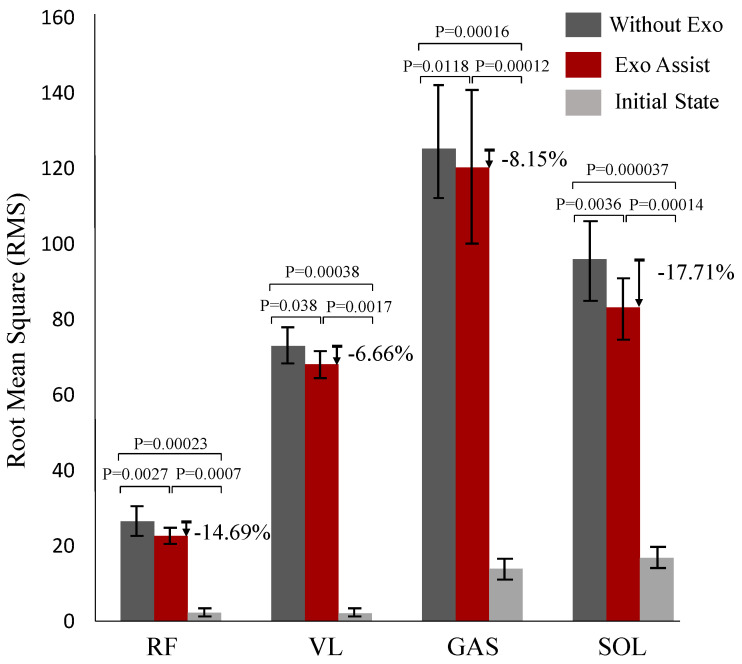
Muscle fatigue calculated by *sEMG* signals. RF, VL, GAS, and SOL represent the rectus femoris, vastus lateralis, gastrocnemius, and soleus, respectively. Exo Assist and Without Exo represents normal walking with the exosuit and normal walking without the exosuit, respectively. The Initial State means the level of muscle fatigue during stand-still.

**Table 1 micromachines-12-01150-t001:** The mass distribution of the lightweight soft exosuit.

Part	Number	Mass (kg)	Location
Waist belt	1	0.26	Waist
Thigh brace	2	0.28	Thigh
Calf brace	2	0.16	Calf
Ankle frame	2	0.12	Ankle
Actuator	1	0.6	Waist
Automatic winder	4	0.22	Thigh/Ankle
Batteries	1	0.53	Waist
IMUs	2	0.046	Thigh
Load cells	2	0.024	Thigh
Total	-	2.24	-

**Table 2 micromachines-12-01150-t002:** The results of muscle fatigue experiments, where NE and AO represent the RAM values of normal walking without using the exosuit and normal walking with using the exosuit, respectively. “Re” represents the reduction.

S	RF			VL			GAS			SOL		
	NE	AO	Re	NE	AO	Re	NE	AO	Re	NE	AO	Re
S1	26.77	22.27	16.81%	72.84	66.56	8.62%	118.68	112.34	5.34%	89.7	74.15	17.33%
S2	24.18	22.32	7.69%	67.62	70.09	−3.65%	120.93	115.22	4.72%	86.52	82.33	4.84%
S3	25.66	22.79	11.18%	71.82	64.88	9.66%	117.62	100.3	14.72%	91.17	71.26	21.83%
S4	25.05	22.5	10.18%	69.81	67.63	3.20%	118.62	101.99	14.02%	93.83	74.39	20.71%
S5	29.07	23.7	18.47%	78.89	71.89	8.87%	142.4	140.39	1.41%	111.99	88.82	20.68%
S6	28.66	22.66	20.93%	77.04	67.8	11.99%	133.66	120.35	9.96%	103.07	83.26	19.22%

**Table 3 micromachines-12-01150-t003:** Comparison of famous soft exoskeletons.

Research	Assistance Mode	Weight (kg)	Power	Number of Motor
Kim et al. [37]	Hip extension	5.004	Powerd	2
Jim et al. [38]	Hip flexion	∖	Powerd	2
Sangjun et al. [39]	Hip extension and flexion & Ankle plantar flexion	5.1	Powerd	4
Collins et al. [23]	Ankle plantar flexion	0.816–1.006	Unpowered	∖
Alan T. Asbeck et al. [17,40]	Hip extension & Ankle plantar flexion	6.2	Powerd	2
Yu et al. [16]	Hip extension & Knee flexion	4.6	Powerd	2
This work	Hip extension	2.24	Powered	1

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
