# Peer review of "A Time Division Multiplexing Inspired Lightweight Soft Exoskeleton for Hip and Ankle Joint Assistance"

_micromachines, 2021, doi:10.3390/mi12101150_

Round 1
Reviewer 1 Report
This is a very interesting study and showed a body-worn time-division multiplexing lightweight soft exoskeleton to provide assistance for hip flexion and ankle flexion. The authors stated the muscle fatigue of rectus femoris, vastus lateralis, gastrocnemius, and soleus decreased by 14.69%, 6.66%, 17.71%, and 8.15%, respectively, comparing to without using the exoskeleton
There are many potential factors that could affect the gait assistance when using the exoskeleton equipment, rather than reducing the mass of the system. The authors should explain this point in detail.
Indeed, early studies showed one motor is required for each joint and helps the patients to drive their movements, but these systems also try to reduce the mass of the system by changing the design of the motor. The authors did review this.
Although the author reviewed researches and pointed out the flexion and extension movements for the joint can be derived or assisted by a single motor, but their limitations should reveal when applying for a walk, such as the accuracy of the joint movement during walking
This study utilizes only one motor to assist hip flexion and ankle plantar flexion of legs, the stability for motor performance and accuracy for the joint movement should claim in this manuscript
The authors used Time-division multiplexing (TMD) to transmit different signals to control multiplex movements, but this should base on both legs is completely symmetry; however, in real-movements for both legs in daily activities (walking, running…), they are still few asymmetries, the authors may explain and represent how they solve this issue.
The authors should explain how much elastic potential energy in J, Nt, or kg was generated during the gait cycle and if it is possible to drive the movements for hip, knee, and ankle joints after the cable is pulled out of the shell
The muscle fatigue test protocols (the walking speed gradually increased from 2km/h to 4.5km/h, rising by 0.5km/h each time and when the speed reached 4.5km/h, the subjects continued walking for 5 minutes. And the sEMG signals were then collected during stable walking at a speed of 4.5km/h) should explain the rationale and based on which researches
The authors may explain why they only choose the muscle fatigue tests to validate and represent the benefits of the lightweight soft exoskeleton.
The authors listed Table 3 to show the effectiveness of the lightweight soft exoskeleton on muscle fatigue; however, the system design, applications, and limitations should be pointed out and discussed in the same paragraph.
Authors concluded that this system is feasible to enhance people’s athletic ability by assisting four joints with only one motor may not appropriate because they only used the muscle fatigue to validate these benefits for the lightweight soft exoskeleton
This study recruited six participants and carried out the muscle fatigue test, but I cannot find any IRB approval (No.??) in this manuscript.
Author Response
Dear reviewer:
First of all, sincerely thank you for your professional review of this manuscript. Next, I will introduce the modified parts.
Question 1:
“There are many potential factors that could affect the gait assistance when using the exoskeleton equipment, rather than reducing the mass of the system. The authors should explain this point in detail.”
Modified content:
Indeed, there are many factors that could affect the effectiveness of assistance, such as supportability, portability, stiffness of system, comfort, biomechanical characteristics of human motions. We add some specific description of these factors form line 24 to 28 and line 31 to line 33. (the words marked in red).
Question 2:
“Although the author reviewed researches and pointed out the flexion and extension movements for the joint can be derived or assisted by a single motor, but their limitations should reveal when applying for a walk, such as the accuracy of the joint movement during walking.”
Modified content:
We add some descriptions to reveal the limitation of using single motor to assist the flexion and extension movements of the joint from 37 to 38 (The words marked in red).
Question 3:
“This study utilizes only one motor to assist hip flexion and ankle plantar flexion of legs, the stability for motor performance and accuracy for the joint movement should claim in this manuscript”
Modified content:
This is a good question. Actually, most of the evaluation methods of the exoskeleton are based on physiological signals such as EMG, which can intuitively see whether the exoskeleton is beneficial to the enhancement of human motor ability. It is also feasible to evaluate the system performance from the stability for motor performance and accuracy for the joint movement. Therefore, we will carry out further research based on this evaluation method.
Question 4:
“The authors used Time-division multiplexing (TMD) to transmit different signals to control multiplex movements, but this should base on both legs is completely symmetry; however, in real-movements for both legs in daily activities (walking, running…), they are still few asymmetries, the authors may explain and represent how they solve this issue.”
Modified content:
The actuation scheme proposed in this paper does not apply to asymmetric gait. In addition, although the biological information of the left and right legs, such as joint torque curve, plantar pressure distribution and so on, may not be exactly equal during walking, the gait phases of both legs are symmetrical for subjects without any musculoskeletal injure or disorder. Meanwhile, this study is currently focusing on the scenario of walking on a treadmill, the assistance to daily walking, often with change in speed and stride length, will be the next step. We add some descriptions from 465 to 469 (The words marked in red). Finally, asymmetric gaits are often seen in people with muscle disorders of the lower extremities, it’s interesting and meaningful to study how to assist asymmetric gaits with our soft exoskeleton and we may focus on this problem in the future work.
Question 5:
“The authors should explain how much elastic potential energy in J, Nt, or kg was generated during the gait cycle and if it is possible to drive the movements for hip, knee, and ankle joints after the cable is pulled out of the shell.”
Modified content:
We add some descriptions about the preload which is produced by automatic winder applied to the force loading path from 151 to 160 (The words marked in red).
Question 6:
“The muscle fatigue test protocols (the walking speed gradually increased from 2km/h to 4.5km/h, rising by 0.5km/h each time and when the speed reached 4.5km/h, the subjects continued walking for 5 minutes. And the sEMG signals were then collected during stable walking at a speed of 4.5km/h) should explain the rationale and based on which researches.”
Modified content:
In our preliminary experiments, we found that due to the sudden acceleration, it took some time for subjects to stabilize their gait, and the time varied from person to person. This affects the accuracy of sEMG data. Therefore, in the final experimental scheme, we gradually increased the speed from 2km/h to 4.5km/h, each time increasing by 0.5km/h, in order to obtain a stable gait at 4.5km/h. We add some descriptions from line 369 to line 371 (The words marked in red).
Question 7:
“The authors may explain why they only choose the muscle fatigue tests to validate and represent the benefits of the lightweight soft exoskeleton.”
Modified content:
The soft exoskeleton proposed in this paper is designed to reduce muscle fatigue of human walking. For this reason, we choose the muscle fatigue tests to validate and represent the benefits of the lightweight soft exoskeleton. Some supplementary explanations are added from 342 to 347. (The words marked in red)
Question 8:
“The authors listed Table 2 to show the effectiveness of the lightweight soft exoskeleton on muscle fatigue; however, the system design, applications, and limitations should be pointed out and discussed in the same paragraph.”
Modified content:
We add some descriptions to relationship between the results of muscle fatigue test, the system design and applications from 392 to 415 (The words marked in red). In addition, the limitations of this soft exoskeleton is described from 461 to 464 (The words marked in red).
Question 9:
“Authors concluded that this system is feasible to enhance people’s athletic ability by assisting four joints with only one motor may not appropriate because they only used the muscle fatigue to validate these benefits for the lightweight soft exoskeleton.”
Modified content:
In fact, human athletic ability needs to be evaluated from many aspects, not just muscle fatigue. What we're trying to express is that reducing muscle fatigue is beneficial to improving people’s athletic ability. The ambiguous statement has been deleted and a more reasonable summary is placed in Section 6, from 483 to 488. (The words marked in red)
Question 10:
“This study recruited six participants and carried out the muscle fatigue test, but I cannot find any IRB approval (No.??) in this manuscript.”
Modified content:
We add some descriptions about the IRB approval from 351 to 355 (The words marked in red).

Reviewer 2 Report
This work presents the design and control of a lightweight soft exoskeleton that assists hip-plantar flexion of both legs in different phases during a gait cycle with only one motor, assists hip flexion and ankle planter flexion on both legs with a single motor which outputs power to the body throughout the gait cycle. The approach is interesting and the results show the efficacy of the proposed method to analyze features designed to assist hip flexion and plantar flexion of both legs with only one motor since there is no overlap between the hip flexion movement and the toe-offs movement of the separate legs during walking. Finally, It is also good to have an analysis and overview review of all new perspectives of soft exoskeletons design, to engage from the entire gait cycle rather than individual joint movements.
To further support this work, it would be important to add the following reference on Soft Exoskeletons in the paragraph: "Therefore, the soft exoskeleton, which avoids the disadvantages of rigid exoskeleton, has attracted the attention of many scholars".
Reference:
Pérez Vidal, A.F.; Rumbo Morales, J.Y.; Ortiz Torres, G.; Sorcia Vázquez, F.d.J.; Cruz Rojas, A.; Brizuela Mendoza, J.A.; Rodríguez Cerda, J.C. Soft Exoskeletons: Development, Requirements, and Challenges of the Last Decade. Actuators 2021, 10, 166. https://doi.org/10.3390/act10070166
After reviewing the manuscript, I have no further comments to make.
In this reviewer's opinion, the work presented is valuable and worthy of being published with minor comments.
Author Response
Dear reviewer:
First of all, sincerely thank you for your professional review of this manuscript. Next, I will introduce the modified parts.
Question 1:
“To further support this work, it would be important to add the following reference on Soft Exoskeletons in the paragraph: ‘Therefore, the soft exoskeleton, which avoids the disadvantages of rigid exoskeleton, has attracted the attention of many scholars’.”
Modified content:
Thank you for recommending such an excellent article. In this article, Various investigations on soft exoskeletons are presented and their functional and structural characteristics are analyzed. We add the reference at line 34 (the words marked in red).

Round 2
Reviewer 1 Report
The authors have replied to the commands very well. After revision, I believe this manuscript can provide evidence-based information with solid descriptions to the readers and be ready to publish.